# Delamination of Novel Carbon Fibre-Based Non-Crimp Fabric-Reinforced Thermoplastic Composites in Mode I: Experimental and Fractographic Analysis

**DOI:** 10.3390/polym15071611

**Published:** 2023-03-23

**Authors:** Muhammad Ameerul Atrash Mohsin, Lorenzo Iannucci, Emile S. Greenhalgh

**Affiliations:** 1Department of Aeronautics, Imperial College London, Exhibition Road, London SW7 2AZ, UK; 2Empa, Swiss Federal Laboratories for Materials Science and Technology, Überland Str. 129, 8600 Dübendorf, Switzerland

**Keywords:** carbon-fibre-reinforced polymer (CFRP), thermoplastic composites, high performance composites, delamination resistance, non-crimp fabric (NCF)

## Abstract

Delamination, a form of composite failure, is a significant concern in laminated composites. The increasing use of out-of-autoclave manufacturing techniques for automotive applications, such as compression moulding and thermoforming, has led to increased interest in understanding the delamination resistance of carbon-fibre-reinforced thermoplastic (CFRTP) composites compared to traditional carbon-fibre-reinforced thermosetting (CFRTS) composites. This study evaluated the mode I (opening) interlaminar fracture toughness of two non-crimp fabric (NCF) biaxial (0/90°) carbon/thermoplastic composite systems: T700/polyamide 6.6 and T700/polyphenylene sulphide. The mode I delamination resistance was determined using the double cantilever beam (DCB) specimen. The results were analysed and the Mode I interlaminar fracture toughness was compared. Additionally, the fractographic analysis (microstructure characterisation) was conducted using a scanning electron microscope (SEM) to examine the failure surface of the specimens.

## 1. Introduction

Delamination is a critical failure mode in composite materials, particularly in laminated composites. These materials are made up of layers of different materials that are bonded together, and delamination occurs when the layers separate from one another. In recent years, there has been extensive research dedicated to enhancing the delamination resistance of composites [1,2,3,4,5,6,7]. The resistance to delamination is often referred to as the Mode I interlaminar fracture toughness and is represented by the $G_{Ic}$ values, which are sometimes quoted in material supplier’s data sheets. However, in the past, most of the research performed in regards to the interlaminar fracture toughness of laminated composites has revolved around thermosetting composites [1,8,9,10,11,12,13,14,15,16]. To the author’s knowledge, very few studies can be found in the open literature on the delamination of laminated non-crimp fabric composite systems [15,17,18,19,20], and these are mostly thermoset systems. Furthermore, most research on the delamination of thermoplastic composites revolved around other types of fibre architecture, e.g., unidirectional laminates [21,22,23].

One of the most useful real-world applications of composites is low-velocity impact, as laminated composites have been proven to be capable of absorbing higher impact energy compared to conventional metals and metal alloys [24,25]. Mohsin et al. [24] have proven that thermoplastic composites are superior to most thermosetting composites. As the delamination of composite structures can be induced by the impact of low velocity, it is one of the major challenges that draws the most attention from a safety perspective. Strength, toughness, and fatigue life are all reduced as a result of the harm brought on by the formation of such a delamination [26]. 

The mode I interlaminar stiffness is often tested and extensively researched using double-cantilever beam experiments [1,7,10,20,27]. To measure the Mode I delamination resistance of a material, the double cantilever beam (DCB) specimen is commonly used. The DCB specimen is a rectangular-shaped piece of material with a notch cut into it, which simulates a crack. The resistance to delamination is measured by loading the specimen until the layers of the composite separate. After the experimental results are collected and data are analysed, the Mode I interlaminar fracture toughness of the material is determined. To understand the cause of failure, the fractographic analysis is conducted using a scanning electron microscope (SEM) to characterise the microstructure of the failed specimens. This allows researchers to understand the underlying mechanisms of delamination and develop strategies to improve the delamination resistance of composites.

The choice of polyamide 6.6 (PA6.6) and polyphenylene sulphide (PPS) as the two matrices presented in this study was driven by the UK-DATACOMP [28] and UK-THERMOCOMP [29] project due to the interest in the UK automotive and aerospace industry. The PA6.6 polymer was considered as a reasonable option for the automotive industry due its relatively low cost. The PPS was of interest due to its typically higher tensile strength and lower processing temperatures (300–345 °C) when compared to other well-known aerospace thermoplastics such as PEEK (350–400 °C) [30,31]. Additionally, this study was aimed at establishing and enhancing the materials database of the industrial partners of the project. Hence, there was a need to characterise the Mode I delamination of these novel NCF biaxial CFRTP systems.

## 2. Materials and Methods

### 2.1. Material System and Laminate Preparation

This research used two types of CFRTP material systems: (i) NCF biaxial (0°/90°) T700 (continuous) carbon fibres pre-impregnated with polyamide 6.6 veils (T700/PA6.6) and PA6.6 stitching, and (ii) NCF biaxial (0°/90°) T700 (continuous) carbon fibres pre-impregnated with polyphenylene sulphide veils (T700/PPS) and Kevlar^®^ (DuPont, Wilmington, DE, USA) stitching. These materials were supplied by partners in the THERMOCOMP project [29]. The T700/PA6.6 material system has also previously been reported and discussed in Mohsin et al. [24,32].

The material (unreinforced and reinforced) and mechanical properties of the constituent materials of the composite are shown in Table 1. However, since the material system is proprietary, the mechanical properties of the laminates were obtained by the author using a series of standardised and non-standardised tests listed in Table 2 and described in [24,33].

Table 1 shows the mechanical properties of neat PA6.6, PPS, and T700 fibre. Table 2 shows the Quasi-static mechanical properties of T700/PA6.6 (FVF = 52%) and T700/PPS (FVF = 61%) [34].

**Table 1 polymers-15-01611-t001:** Mechanical properties of neat PA6.6, PPS, and T700 fibre.

	PA6.6	PPS	T700 Fibre
References	[33,35,36]	[33,37,38]	[39]
Density, ρ (kg/m^3^)	1170	1310	1800
Tensile strength, ultimate (MPa)	71	111	4900
Tensile modulus (GPa)	0.2–3.8 *	2.6–6.1 *	230
Elongation at break (%)	53.9	13.9	2.1
Mode I fracture toughness, $G_{IC}$ (kJ/m^2^)	0.2	0.5	-

* Note that the mechanical properties of the polymers were not characterised/measured in-house. Based on vendor’s data and the grade of the of matrix, the estimated tensile modulus of the PA6.6 and PPS are ~3.5 GPa and ~3 GPa, respectively.

**Table 2 polymers-15-01611-t002:** Quasi-static mechanical properties of T700/PA6.6 (FVF = 52%) and T700/PPS (FVF = 61%) [24,33,34,40].

Mechanical Properties	Material: T700/PA6.6	T700/PPS
Tensile Young’s modulus (GPa)	65	60 ^1^
Compressive Young’s modulus (GPa)	69	47 ^1^
Tensile strength (MPa)	918	852
Compressive strength (MPa)	461	265
In-plane shear modulus (GPa)	3.2	3.3
In-plane shear stress at 5% (MPa)	52	73

^1^ The measured tensile and compression Young’s modulus of the T700/PPS, which has a higher fibre volume fraction (61%) than the T700/PA6.6 (52%), is lower due to the influence of the stitching material of the former (Kevlar^®^), causing undulation in the laminate. This reduces the modulus.

To ensure accuracy and eliminate any effects of moisture, the densities of both CFRTP material systems were measured using a pycnometer after being stored in an oven at 40 °C for three days. The densities of the T700/PA6.6 and T700/PPS systems were measured to be 1485 kg/m^3^ and 1553 kg/m^3^, respectively.

Additionally, the fibre–volume–fraction (FVF) of the laminates produced was measured via a thermogravimetric analysis (TGA) (T700/PA6.6 = 52% and T700/PPS = 61%). The process has been detailed in [34]. The FVF measurement is important to understand the composite material properties and its behaviour under a load. The accurate measurement of FVF is essential to make sure that the design is robust enough and to select the appropriate composite material for a specific application.

### 2.2. Manufacturing Process

The laminates were created by layering T700/PA6.6 with a layup sequence of (0/90)_12s_ using a hand lay-up method. The hand lay-up method is a process where the layers of material are manually placed and arranged on the mould before being shaped with a thermoforming method. The thermoforming method used in this case was a laboratory hydraulic press at 275 °C. This process of layering and shaping the T700/PA6.6 material creates a laminate that is strong and durable.

The recommended processing parameters for T700/PA6.6 are:Dwell time: 10 min;Processing temperature: 275 °C;Heating rate: 15 °C/min;Pressure: 1.5 MPa;Demoulding temperature: 25–35 °C.

The average thickness of the T700/PA6.6 panel was measured to be 4.1 ± 0.28 mm. This measurement is important, as it ensures that the final product meets the desired thickness specifications.

For the T700/PPS, the manufacturer’s recommended processing parameters are as follows:Dwell time: 10 min;Processing temperature: 315 °C;Heating rate: 15 °C/min;Pressure: 2.5 MPa;Demoulding temperature: <100 °C.

These parameters are similar to the ones for T700/PA6.6, with the main difference being the processing temperature of 315 °C and the pressure of 2.5 MPa. The demoulding temperature is also slightly different, with T700/PPS having a more flexible range of simply < 100 °C.

In summary, the laminates were prepared by layering and shaping T700/PA6.6 and T700/PPS materials using a hand lay-up method and a laboratory hydraulic press, with specific processing parameters to ensure the optimal bonding and curing of the material. The final product meets the desired thickness specifications and has a good strength and durability.

## 3. Experimental Setup and Data Reduction

### 3.1. Experimental Setup

Measuring a laminated composite material’s resistance against the interlaminar fracture, known as the Mode I interlaminar fracture toughness ($G_{Ic}$), is critical in the process of material selection and design for various structural applications. The reason for this is that vulnerability to delamination has always been considered as one of the most vital weaknesses of advanced laminated composite structures. Therefore, the knowledge of a laminated composite material’s resistance against the interlaminar fracture is always valuable in the process of material selection and design for various structural applications.

One of the most commonly used methods to determine $G_{Ic}$ values is the DCB test. The DCB test allows for the determination of the Mode I interlaminar fracture toughness (critical energy release rate), $G_{Ic}$. The DCB test performed on both NCF biaxial carbon-fibre-reinforced thermoplastic (CFRTP) systems (T700/polyamide 6.6 (PA6.6) and T700/polyphenylene sulphide (PPS)) in this study is partially in accordance to the existing standardised test outlined in the American Society for Testing and Materials (ASTM) D5528-13 [41], which was originally tailored for unidirectional fibre-reinforced plastic composites.

The advantage of using $G_{Ic}$ values is that they are independent of the specimen or method of load introduction. This makes $G_{Ic}$ values useful for determining the design allowable and damage tolerance, specifically, the delamination failure criteria of composite structures manufactured from a specific composite material system. This makes the DCB test a valuable tool for engineers and designers in order to select the appropriate composite material for a specific application and to make sure that the design is robust enough to avoid a delamination failure.

The typical the DCB specimen types are shown in Figure 1. It is important to note that the specification of the specimen dimensions and the preparation of the specimen are crucial for the accuracy of the results obtained from the DCB test. Therefore, it is recommended to follow the standardized procedures outlined in ASTM D5528-13 in order to obtain accurate results.

The test specimen is a rectangular laminated composite material with a length of 170 mm and a width of 20 mm. It should have a uniform thickness of 4 ± 0.2 mm and a 60 mm non-adhesive insert in the middle that serves as the point of delamination initiation (Figure 2). The specimen is attached to loading blocks or hinges, which are bonded to one end of the specimen using an epoxy glue, and opening forces are applied to these points. The width and thickness of each specimen were measured and averaged across its length. During the test, the specimen is opened by controlling the opening displacement while measuring the delamination length and load. The DCB specimens were coated in white before the test to improve visibility (Figure 3) and the experimental setup of the test can be observed in Figure 4, which includes a travelling microscope used to record the delamination length and monitor crack growth. The test machine used is an INSTRON^®^ testing machine (Instron Corporation, Norwood, MA, USA) and the cross-head displacement rate for each test was set to 0.5 mm/min.

### 3.2. Data Reduction

The graph of load versus cross-head displacement is generated by plotting the data obtained from the test. The Mode I interlaminar fracture toughness is then calculated using the modified beam theory (MBT) method [42]. The Mode I interlaminar fracture toughness, also known as the strain energy release rate, is a measure of the resistance of the material to the crack propagation and is represented by $G_{Ic}$. This value can be calculated using Equation (1).
(1)$$G_{Ic}=\frac{3P\delta }{2b\left(a+\left|\Delta \right|\right)}$$
where *P* is the load, *δ* is the cross-head displacement, *b* is the specimen width, *a* is the delamination length, and Δ is a correction term applied to the delamination length. 

The correction term, Δ, is determined from the experimental data by generating a least square plot of the cubic root of compliance, *C*^1/3^, as a function of delamination length, *a*. This is conducted to account for any variations in the delamination length that may have occurred during the test. The correction term, Δ, is the value that is added to the delamination length to make the plot pass through the origin. The compliance, *C*, can be calculated using the Equation (2).
(2)$$C=\frac{\delta }{P}$$

This equation is used to determine the relationship between the displacement and the load applied on the specimen. The results of these calculations are then used to determine the mode I interlaminar fracture toughness of the material.

## 4. Results and Discussion

The DCB test, which was performed on both T700/PA6.6 and T700/PPS specimens, revealed a stable crack growth throughout the duration of the test. The failure was determined by the first significant load drop in the load-displacement curve, as outlined in the appendices. During the test, it was observed that the crack propagated consistently along the midplane and length of the specimen, as the load gradually decreased with the increasing delamination length. The maximum delamination lengths of the T700/PA6.6 samples, before the DCB arms exhibited a bending failure, were found to be larger, at around 70–80 mm compared to the 40–50 mm observed in the T700/PPS samples.

Additionally, the R-curves for both specimens showed a positive slope, which is a common characteristic for these types of materials. This can be observed in Figure 5 and Figure 6 for the T700/PA6.6 and T700/PPS samples, respectively. Furthermore, a representative R-curve is highlighted against the crack length in Figure 7, providing a clear visual representation of the results.

Based on the R-curves obtained for both specimens, the Mode I fracture toughness, $G_{Ic}$, for both samples, was calculated and tabulated as shown in Table 3. The average $G_{Ic}$ calculated for the T700/PA6.6 and T700/PPS were 1.50 kJ/m^2^ (CV = 9.7%) and 1.75 kJ/m^2^ (CV = 8.6%), respectively. These results demonstrate that the Mode I interlaminar fracture toughness of the T700/PPS is approximately 17% higher than the T700/PA6.6 system.

In conclusion, the DCB test provided valuable information on the stable crack growth and delamination behaviour of the T700/PA6.6 and T700/PPS specimens. The results of the test, including the R-curves, the maximum delamination lengths, and the Mode I fracture toughness, have allowed for a comprehensive understanding of the mechanical properties of these materials. These findings can be used to inform future materials’ development and application decisions.

## 5. Fractographic Analysis

Fractographic analysis is a powerful tool that allows for the characterisation of the microstructure of materials, particularly in the case of delamination failures. By using a scanning electron microscope (SEM), the morphologies of the fractured surfaces can be examined and interpreted as a Mode I fracture. Mode I fractures are characterised by rough surfaces, due to the presence of broken fibre ends, and can be further divided into two types: matrix cleavage and fibre bridging [47].

At the microstructural level, the delamination failure extends along the fibres and propagates into the surrounding matrix, as shown in Figure 8 and Figure 9. The textured micro flow and corresponding river lines converge within the matrix next to the fibres, and are the key features of the matrix cleavage. The river lines’ growth direction typically follows the direction of the global crack growth [11].

The degree of fibre bridging is often associated with the material and processing conditions of the laminated composites, and it tends to increase with increasing crack lengths [47]. Additionally, it can also contribute to a corrugated cross-section in the fracture surface, as shown in Figure 9.

The corrugated cross-section of the T700/PPS samples after the test, as shown in Figure 9, is the result of extensive fibre bridging and a combination of fibre-matrix debonding, matrix deformation, and void formation. This also contributes to the stick-slip crack growth behaviour that was observed during the experiment. The distinctively associated fracture morphology with thermoplastic composites, namely ductile drawing, could be observed in both systems, as shown in the micrographs in Figure 8 and Figure 9.

These results were largely influenced by the through-thickness stitching of the NCF. The presence of stitching tends to enhance the Mode I delamination resistance. Based on Figure 10, the stitches of the T700/PPS are more apparent on the fracture surfaces compared to the T700/PA6.6. The stronger Kevlar^®^ stitches on the former (compared to the Nylon^®^ (DuPont, Wilmington, DE, USA) (stitching on the latter), to a certain extent, promote a greater crimp development of the tows. This results in an increase in the fracture toughness. Additionally, in materials with a high degree of crimp, stitches themselves can sometimes fail. When the stitches impose too much or add insufficient constraint to the tows, fibre waviness can develop. This misalignment leads to an increase in Mode I toughness and reduction in compression strength.

## 6. Conclusions

The results of the study indicate that the performance of the composite laminate systems is highly dependent on the stitching material and the mechanical properties of the polymer used. This was evident from the microstructure characterisation of both systems, which was conducted using fractography. The analysis revealed that the crack propagation characteristics of both laminate systems were very comparable, with both systems displaying stable crack growth. Additionally, both systems showed stable delamination growth, which is an important factor to consider when evaluating the overall strength and durability of the composite materials.

One of the key findings of the study was that both CFRTP systems performed superiorly when compared to typical CFRTS systems. This is likely due to the fact that the CFRTP systems are able to maintain their structural integrity better under high stress and strain conditions, which is a result of the improved mechanical properties of the polymer used. This makes the CFRTP systems more suitable for applications that require high strength and durability, such as in aerospace and automotive industries.

Overall, the study provides valuable insights into the performance of composite laminate systems and the factors that influence their properties. The findings of the study can be used to develop more advanced composite materials that can meet the demanding requirements of various industries.

## Figures and Tables

**Figure 1 polymers-15-01611-f001:**
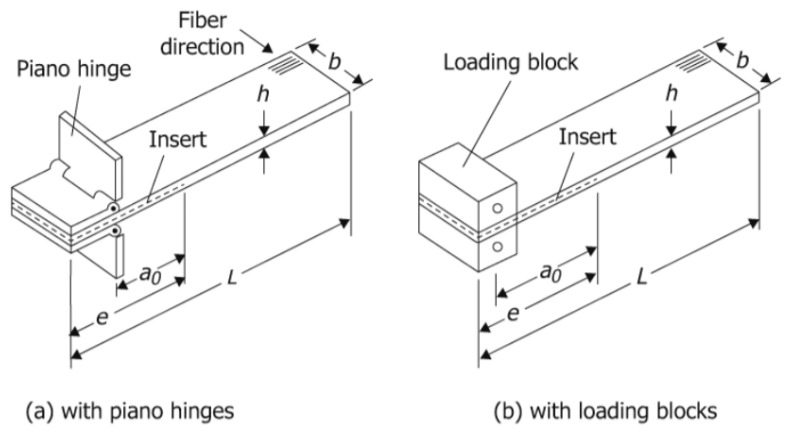
Standardised DCB specimens according to the ASTM D5528-13 [41]: (**a**) with piano hinges and (**b**) with loading blocks. Reprinted, with permission, from ASTM D5528-13 [41] Standard Test Method for Mode I Interlaminar Fracture Toughness of Unidirectional Fibre-Reinforced Polymer Matrix Composites, copyright ASTM International. A copy of the complete standard may be obtained from www.astm.org.

**Figure 2 polymers-15-01611-f002:**
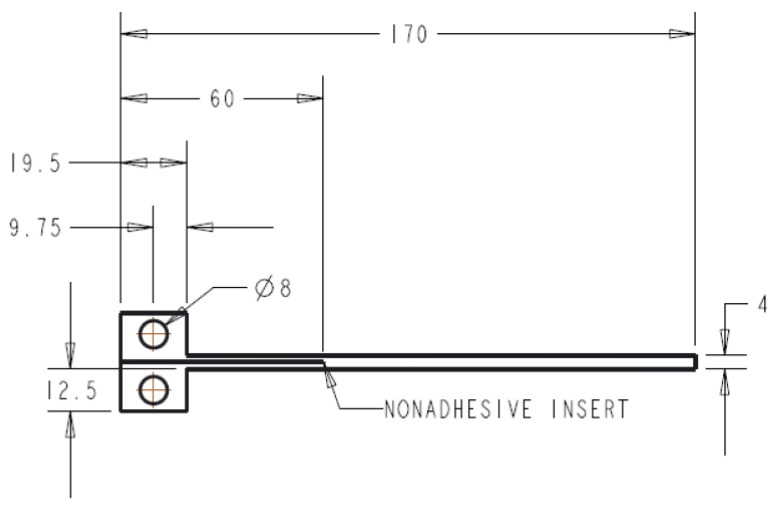
Exact dimensions of the DCB specimen (in mm).

**Figure 3 polymers-15-01611-f003:**
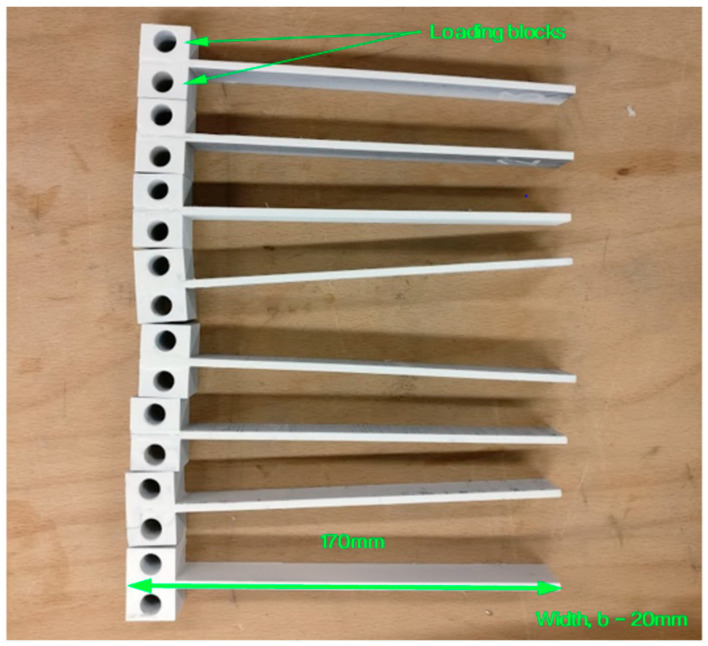
The DCB specimens that have been painted white prior to being marked to enhance visibility.

**Figure 4 polymers-15-01611-f004:**
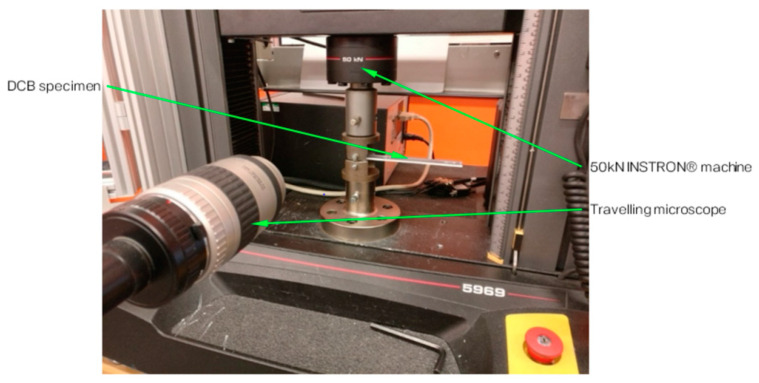
DCB test setup with a travelling microscope, which is a microscope that is mounted on a slider that can be moved along a scale.

**Figure 5 polymers-15-01611-f005:**
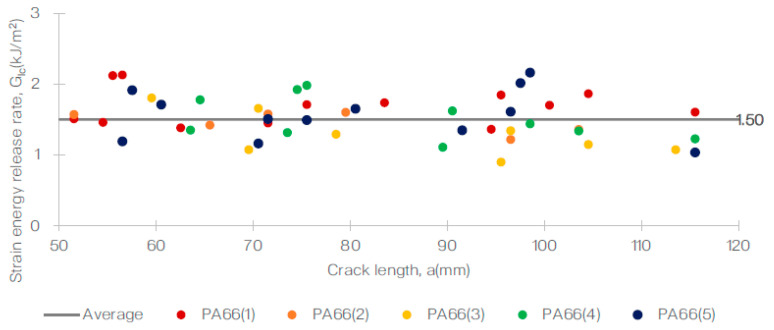
Delamination resistance curve (R-curve) from the DCB test of the T700/PA6.6 system.

**Figure 6 polymers-15-01611-f006:**
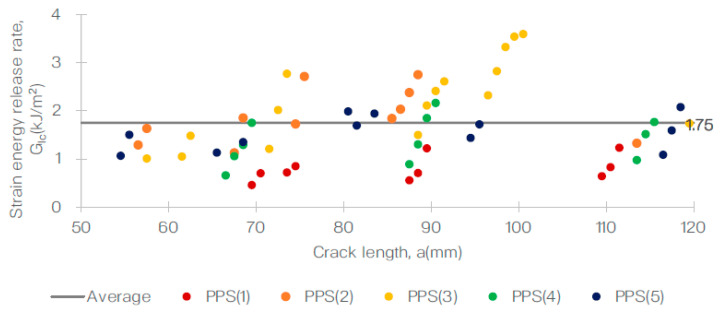
Delamination resistance curve (R-curve) from the DCB test of the T700/PPS system.

**Figure 7 polymers-15-01611-f007:**
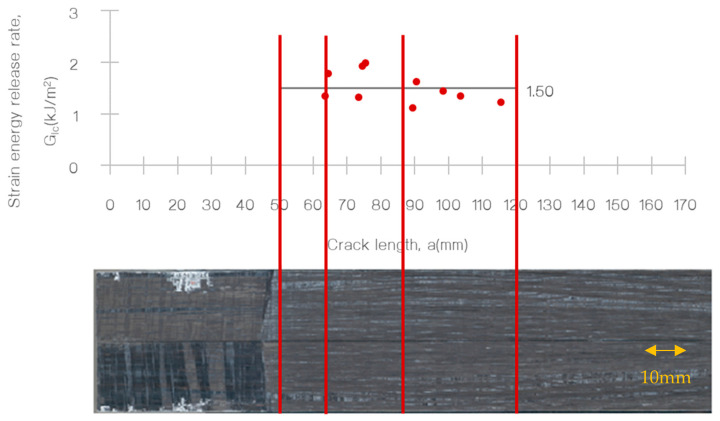
Representative R-curve (specimen #1 T700/PA6.6) against measured crack length.

**Figure 8 polymers-15-01611-f008:**
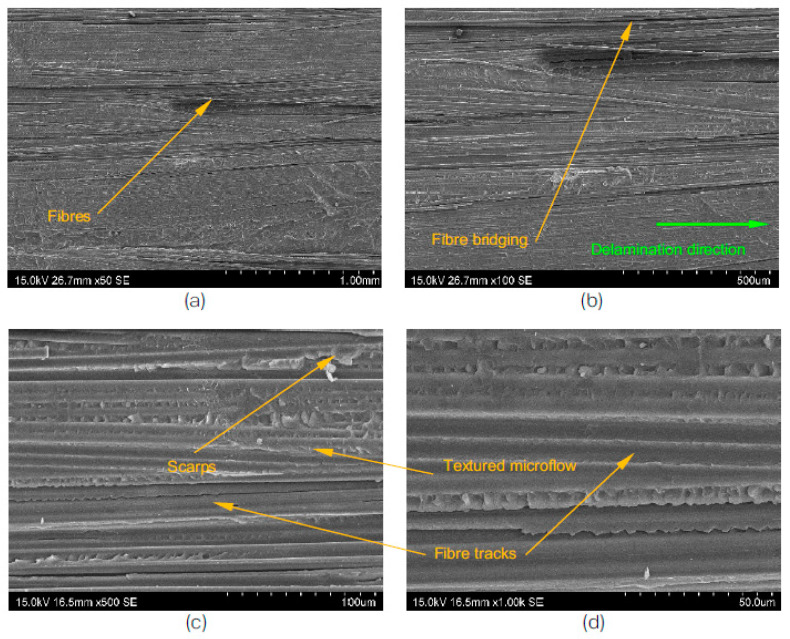
SEM view of the T700/PA6.6 DCB specimens after the test; scale bar is (**a**) 1 mm, (**b**) 500 μm, (**c**) 100 μm, and (**d**) 50 μm.

**Figure 9 polymers-15-01611-f009:**
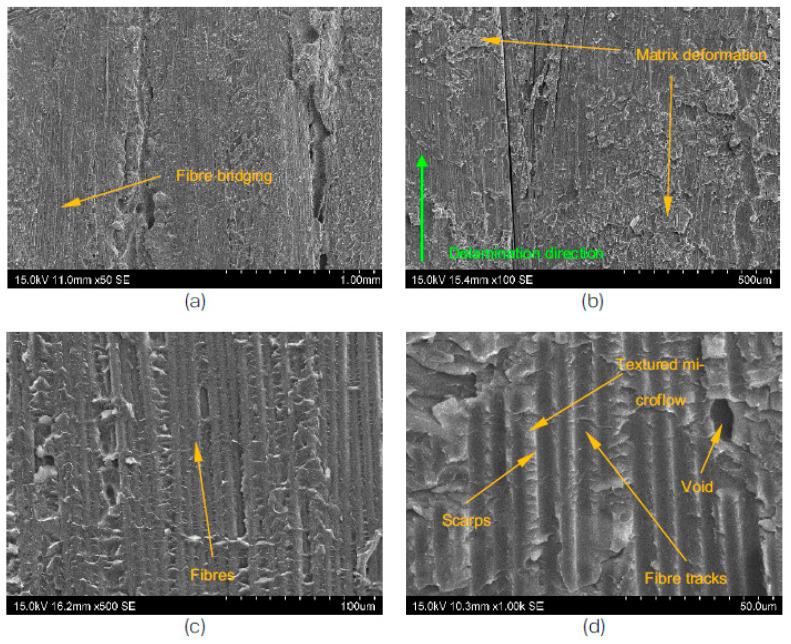
SEM view of the T700/PPS DCB specimens after the test; scale bar is (**a**) 1 mm, (**b**) 500 μm, (**c**) 100 μm, and (**d**) 50 μm.

**Figure 10 polymers-15-01611-f010:**
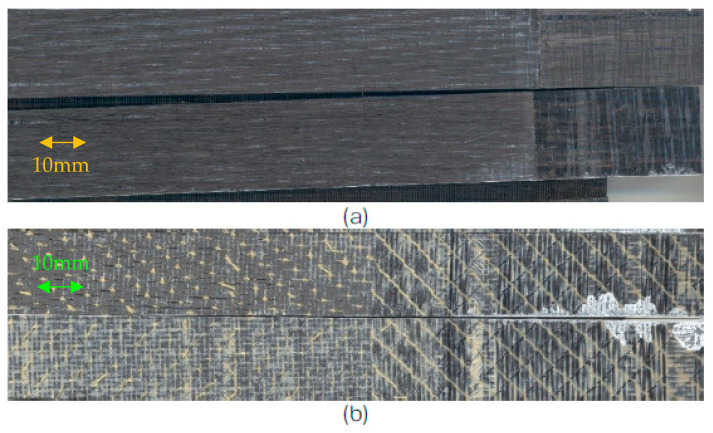
Mode I failure surfaces of NCF (**a**) T700/PA6.6 and (**b**) T700/PPS; crack growth from right to left.

**Table 3 polymers-15-01611-t003:** Summary of the Mode I interlaminar fracture toughness of T700/PA6.6 and T700/PPS.

Study	Material System	Fibre Orient.	$G_{Ic}\;(\mathrm{kJ}/m^{2})$	CV (%)
Mohsin et al. [33]	T700/PA6.6 (this study)	Bidirectional ^1^	1.50	9.7
Mohsin et al. [33]	T700/PPS (this study)	Bidirectional ^1^	1.75	8.6
Pret et al. [43]	AS4/PEEK	Unidirectional	1.46	
Ivanov et al. [44]	T300JB/PPS	Multidirectional ^2^	0.81	
Ivanov et al. [44]	T300/BP-907	Multidirectional ^2^	1.29	
Tijs et al. [45]	AS4D/PEKK	Unidirectional	0.70	
Reis et al. [46]	HS-Carbon/PA6	Unidirectional	2.20	

^1^ NCF biaxial 0/90°; ^2^ Woven 0/90°.

## Data Availability

Not applicable.

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
