# Peer review of "Delamination of Novel Carbon Fibre-Based Non-Crimp Fabric-Reinforced Thermoplastic Composites in Mode I: Experimental and Fractographic Analysis"

_polymers, 2023, doi:10.3390/polym15071611_

Round 1

Reviewer 1 Report

This study aims to apply experimental and fractographic analyses to investigate the Mode I delamination of carbon-fibre non-crimp fabric reinforced thermoplastic composites. However, several issues must be well addressed before acceptance for publication. The reviewer’s comments are as follows:

(1)     What is the biggest contribution of this study? The reviewer cannot see the significance of this study which must be well presented.

(2)     Section 1.1 should not be presented in Introduction, which must be moved into somewhere for “Experimental setup”.

(3)     Almost all figures, such as Figure 1, 2, etc. need to be clearly redrawn with high-resolution. The formats of all texts and numbers in figures must be consistent. For example, the texts in Figure 3, etc. are unclear with low-quality.

(4)     What is the “unit” in Figure 2?

(5)     In Figure 4, what is a “travelling microscope”? How it works? What is the maximum magnification it can achieve?

(6)     Please add scale bar for Figure 7, 10, etc.

(7)     The “fibre bridging” pointed out in Figure 8 is confusing, please add more details or magnified views to strengthen the description.

(8)     In Figure 7 and some figures, the texts, points, and other lines are crossing and covered with each other, which should be modified.

(9)     To improve the significance of this study, the authors should mention some practical cases in relation to “delamination” in Introduction, e.g. delamination in composite panels: Compos Struct 2017; 168: 322-334, Int J Mech Sci 2019; 164: 105160, Int J lightweight mater 2023; 6: 108-116, delamination in composite tubes: Int J Crashworthiness 2021; 26(5): 526-536.

Author Response

(1) The contribution of this study is the mode I interlaminar fracture characterisation of a novel NCF-based CFRP with different stitching materials

(2) I have relocated this to the experimental setup section

(3) I have updated the figures

(4) I have added the unit in the figure caption, in mm

(5) I have added the description of "travelling microscope" at the end of the figure caption in Figure 4

(6) I have added a scale in Figure 7, 10

(7) In Figure 8(b), it can clearly be seen that the fibres are protruding out-of-plane, which is this case, indicates fibre bridge. Unfortunately, I no longer have access to the specimen and therefore, could not provide a new micrograph from a different angle.

(8) I have updated Figure 7 to a higher resolution format. The lines are meant to cross the text and the other lines to correlate the length to the sample

(9) I have added the reference of some of the papers suggested

Reviewer 2 Report

1. Line 43: What do you mean “data is reduced”?

2. Table 1 and Table 2: PPS resin itself has a higher modulus than that of PA66 and T700/PPS also has a higher FVF than that of T700/PA6.6. Then why T700/PPS shows lower modulus than T700/PA6.6 in Table 2?

3. The Introduction section of this paper is very weak and definitely needs to be revised thoroughly. Research background, objective, novelty, and significance of this work need to be emphasized in the introduction section. Many questions need to be answered, including but not limited to: why PA6.6 and PPS were chosen as matrix here? Why not choose other thermoplastics? What’s their targeted application? Authors need to strengthen this section. Here is a most recent good paper for your reference (Qihui Chen et al. Composites Part B: Engineering (2023): 110503).

Author Response

Hi,

Thank you for your comment and feedback. They are highly appreciated. I have since made the following changes to the paper:

  1. Sorry, I meant to say that the data is 'analysed' not 'reduced'
  2. Yes, I understand. I think what I should indicated the range of the Young's modulus of both PA6.6 and PPS in Table 1—I've done exactly that. The modulus of PA6.6 ranges from 0.2-3.8GPa, and the modulus of PPS ranges from 2.6-6.1. As I have not measured the properties of the neat polymer, we can't know for sure, the exact modulus of each system. The stiffest PA6.6 could outperform the least stiff PPS. So, we can't apply rule of mixture here. Nevertheless, Table 2 represents the measured value of both composite systems.
  3. I have expanded the introduction and elaborated on the significance of the study, referred to the paper suggested (thank you).

Round 2

Reviewer 1 Report

The comments have been basically addressed.

Author Response

Thank you for your feedback. 

I have added a few more references and statements to strengthen the paper.

Kind regards.

Reviewer 2 Report

The manuscript has been improved a lot. There are still some comments for you to strengthen this work for publication:

If you know the grades of PA66 and PPS, you should get an estimate of their modulus from vendor’s technical specification data. So that you can verify if your data make sense or not.

Line 122: references needed to support your statement on PEEK processing. (Journal of Applied Polymer Science 137.33 (2020): 48966; Polymer 160 (2019): 231-237; Polymer International 70.8 (2021): 1090-1098; Journal of Applied Polymer Science 139.33 (2022): e52784.)

Author Response

Hi,

Thank you again for your feedback.

I have added two footnotes (lines 86-88 and lines 91-93) to expand the properties of the resin further as well as decribe the perceived lower modulus of the T700/PPS.

I have also made reference to two of the papers suggested.

Thank you.
